# Determinants of clinician and patient to prescription of antimicrobials: Case of Mulanje, Southern Malawi

**Morris Chalusa**[1,2]*, **Felix Khuluza**[3], **Chiwoza Bandawe**[4]

**1** Pathology Department, College of Medicine, University of Malawi, Blantyre, Malawi, **2** Mulanje District Hospital, Ministry of Health, Lilongwe, Malawi, **3** Pharmacy Department, College of Medicine, University of Malawi, Blantyre, Malawi, **4** Department of Mental Health, College of Medicine, University of Malawi, Blantyre, Malawi

* morrischalusa@yahoo.com

**Data Availability Statement:** The datasets used and analyzed during the current study are available under supplementary information.

## Abstract

### Background

Antimicrobial resistance is an emerging problem in low- and middle-income countries. The problem is exacerbated by inappropriate prescription of antimicrobials. Factors that lead to overuse or inappropriate prescription of antimicrobials by the cadre of medical assistants, clinical technicians and clinical officers have received limited attention. This study investigated factors that influence prescription behaviours of antimicrobials among clinical officers in various health facilities in Mulanje district, Southern Malawi.

### Methods

Qualitative study design exploring determinants of antimicrobial prescription from May to October, 2019, was used. In-depth interviews (n = 18) and focus group discussions (n = 2) were conducted with medical assistant (MA), clinical technicians and clinical officers (CO) from four health facilities in Mulanje district. COs are licensed medical practitioners with an initial three-year training and one-year internship while MAs are licensed medical practitioners with initial two-year training and one year internship. Purposive sampling was done to arrive at a sample size of 30 health cadres.

### Results

Participants pointed out that patient preferences, beliefs and clinicians' inadequate education on this issue were among the factors that contribute to inappropriate antimicrobial prescription. 75% of clinicians showed lack of knowledge on the definition of antibiotic and antimicrobial resistance.

### Conclusion

Inappropriate use of antimicrobials is facilitated by prescription decisions made by clinicians who are greatly influenced by their patients. Interventions aimed at improving antimicrobial prescription should target both clinicians and patients.

**Funding:** This study was funded by the Norwegian Research and Capacity Building for Higher Education (NORHED) Antimicrobial Stewardship project, grant number QZA-0484RSA-13/0010 to MC. The funders had no role in the design of the study and data collection, analysis, interpretation of data and in writing the manuscript.

**Competing interests:** The authors declare that no conflicts of interest exist.

**Abbreviations:** AMR, Antimicrobial resistant; ABR, Antibiotic resistant; LMICs, Low Middle-Income Countries.

## Introduction

Antimicrobial resistance (AMR) is a great public health challenge that is accelerated by the inappropriate use of antimicrobials [1]. Overprescription of antimicrobials is associated with increased risks of prolonged hospital stay, self-medication of self-limiting conditions, amplified frequent admission to hospital and causing severe infections [2]. Globally, at least 700,000 people die each year of drug resistance illnesses because of infections such as bacteria, malaria, tuberculosis and HIV / AIDS [3]. Antibiotics are used both in humans and in animal health care, and this has also resulted in resistant bacteria emitted into the environment through manure and sewage [4].

A study conducted in Malawi reported that there is a decrease in bacterial bloodstream infection which has been accompanied by a rise in antimicrobial resistance involving all bacterial bloodstream infection pathogens [5]. A recent study done in Malawi found that Gram-positive pathogens are resistant to empiric, first-line antimicrobials [6]. According to GLASS report, 2014, some microorganisms are resistance to some antibiotics. For instance, Escherichia coli are resistance to 3rd generation of cephalosporins and fluoroquinolones, Klebsiella pneumoniae are resistance to 3rd generation of Cephalosporins and 3rd Carbapenems, Staphylococcus aureus are not susceptible to methicillin (MRSA), Nontyphoidal Salmonella are not susceptible to fluoroquinolones, Neisseria gonorrhea are resistance to 3rd generation cephalosporins and finally, Streptococcus pneumonia are non-susceptible or resistant to penicillin [7].

Patient pressure and customer satisfaction are considered to be major factors for inappropriate antibiotic prescription [8]. In sub-Saharan settings like Malawi, clinical officers (COs) are tasked with the prescription of medicines to patients. COs are licensed medical practitioners with an initial three-year training and one-year internship while medical assistants are licensed medical practitioners with initial two year training and one year internship [9]. According to Mangione (2019), clinicians are more likely to prescribe antibiotics if they perceive that the parents who have brought sick children to the hospital want antibiotics if they ask about the treatment plan [8].

In Africa, factors that influence the prescribing of antibiotics by general practitioners (GPs) include patients' demands for antibiotics, prescribing antibiotics to save time due to the perception that it takes longer to explain why antibiotics are not needed, concerns that the patient may not return for follow up, uncertainty in the diagnosis where antibiotics may be warranted, concerns about possible complications, preservation of the doctor-patient relationship, and knowledge and attitudes to AMR [10, 11].

However, improvements in malaria diagnostic tests have promoted the prescription of antibiotics. Many patients who test negative for malaria are treated with antibiotics indiscriminately [12]. This is where the focus on prescribing habits of clinicians has to be looked at as it may inadvertently contribute to AMR.

Szymczak explained that clinicians identify patients' pressure, expectation and demand as a major barrier to more judicious prescribing in particular where a shortage of consulting time meant that the doctor felt unable to adequately explain why antibiotics were inappropriate [13, 14]. Brookes-Howell, et al. described how clinicians spoke of familiarity with the patient, which helped clinicians in their decision on whether to prescribe antibiotics or not [15].

The basis of antibiotic misuse that have been reported so far are; the Physicians' express desire for a quick fix, the problem of diagnostic uncertainty, time-consuming and unrewarding in explaining why antibiotics are not necessary, and lastly patients' lack of knowledge [16].

Among all these, communication skills and diagnostic uncertainty rank among the principal indirect factors influencing antibiotic prescription [17]. In a study conducted in Malaysia, the majority of the respondents agreed that excessive antibiotic prescriptions, using too many

broad-spectrum antibiotics and excessive use of antibiotics in livestock were leading contributors to AMR. In the same study, another group felt that long duration of antibiotic treatment, low dosing of antibiotics, poor hand hygiene and not removing the focus of infection are among the major factors contributing to AMR [18].

Since a previous study on antimicrobial prescriptions focused on physicians and medical doctors, it was necessary to also get views from clinical officers, clinical technicians and medical assistants who prescribe and provide an important amount of the antibiotics prescribed, particularly in Sub-Saharan Africa.

Antimicrobial prescription can also be reduced in setting where there is one-on-one patient-directed education in the workplace [19]. The present paper was trying to understand the prescription patterns that can drive AMR from clinical officers, clinical technicians and medical assistants, particularly in Sub Saharan Africa. These health practitioners are the one at the frontline in providing health services in primary and secondary health care settings in this region [20]. They carry out a range of duties including outpatient department care, emergency care, ward management of hospitalized patients, surgical care and prescriptions of antimicrobials.

The total area of the Mulanje district is 2,056 square kilometres and the current population is estimated at 587,553. The district has two hospitals and 21 health centers and more than 70 clinicians. It is a border area between Malawi and Mozambique and it has also registered high prevalence rate of HIV which is currently 20.6% [21].

## Methods

### Study design, sample size and recruitment of participants

This was a qualitative study aimed at exploring clinicians' views and experiences about prescribing antimicrobials and AMRs. The research question we tackled was: 'What factors determine the Malawian clinician's decision to inappropriately prescribe antimicrobials?' In-depth interviews and focus group discussions were used. They were chosen because they provide much more detailed information, and they allow for a more relaxed atmosphere. They were also chosen because responses can be clarified and expanded upon with probing question. Additionally, interviewees can react and build upon each other's response to provide information or ideas that, they might not have individually provided.

Clinicians from two hospitals, and two health centers in Mulanje district were recruited. There were 30 clinicians who took part in the study. Purposive sampling was utilised. This is appropriate in qualitative research, where the aim is not to obtain a statistically representative sample and make statistical inferences from the results, but rather to obtain an information rich sample and make logical inferences from that sample [22].

We collected data between May and October 2019. Clinicians' qualifications ranged from certificate, diploma to degree in clinical medicine.

### Ethical considerations

Ethical approval was sought from Malawi College of Medicine Research Ethics Committee and granted with approval number P.04/19/2656. Administrative clearance was obtained from the District Medical Officer of Mulanje District Hospital and Mulanje Mission Hospital. Recruited clinicians were given a consent form to read and sign, once they had agreed to take part in the study. All participants were informed about the objective of the study and they enrollment was voluntary. Clinician were assured of privacy and confidentiality. However, only the team that was involved in data collection had access to the information andcodes were used instead of names. Clinicians were also informed on free withdrawal at any time.

Study Participants signed an informed consent form to indicate their willingness to participate in the study.

## Data analysis

Analysis began soon after data collection to get familiarization. It involved reading the transcripts repeatedly and noting down ideas. The information which is pertinent to participants' determinants of antimicrobial prescriptions was identified and coded based on a deductive and inductive approach [23]. These codes were collected into sub themes and themes. The first author subsequently discussed the coding, sub-themes and themes with two independent researchers to enhance data reliability. Themes were reviewed by co-authors (CB and FK). Discrepancies were resolved by reaching a consensus. The first author then presented the findings to the study participants and obtained their feedback to ensure that their perspectives were accurately and clearly represented.

## Results

### Participants' demographic characteristics

There were 17 male and 13 female participants. Their educational backgrounds ranged from medical assistant (certificate in clinical medicine), clinical technician (diploma in clinical medicine) and clinical officer (degree in clinical medicine). All participants (n = 30) reported that they had prescribed antimicrobials in the previous year (Table 1).

### Main themes

Seven main themes influencing antimicrobial prescribing emerged from the semi-structured interviews. Quotations from the interviews are included where relevant to illustrate a point.

Fig 1: Maps of sub themes, themes on the determinants of antimicrobial prescription that emerged from the semi-structured interviews.

### Theme 1: Patients' preferences

The interviewer asked them about determinants of antimicrobial prescription in health care settings. Most clinicians mentioned that some patients force clinicians to prescribe antimicrobials and others come with their own diagnosis to the hospital. The clinicians added that a patient's signs and symptoms and a patient's preference for antimicrobials are significant determinants of antimicrobial prescriptions. Most clinicians reported that patient's preference on antimicrobials was a factor that contributed to inappropriate prescription of antimicrobials.

**Table 1. Demographic characteristic of clinicians who participated in the study at chonde health center, chambe health center, Mulanje Mission Hospital and Mulanje District Hospital.**

| Variable | Category | Medical assistants | Clinical technicians | Clinical Officers | Total |
|---|---|---|---|---|---|
| Sex | Male | 4(13.3%) | 12(40.0%) | 1(3.3%) | 17(56.7%) |
| | Female | 7(23.3%) | 4(13.3%) | 2(6.7%) | 13(43.3%) |
| Facility | MMH | 0(0.00%) | 7(23.3%) | 2(6.7%) | 9(30.0%) |
| | MJDH | 11(36.4%) | 9(29.4%) | 1(3.3%) | 21(70.0%) |
| Professional qualifications | Certificate | 11(36.7% | 0 (0.0%) | 0(0.0%) | 11(36.7%) |
| | Diploma | ) 0(0.0%) | 16(53.3%) | 0(0.0%) | 16(53.3%) |
| | Degree | 0(0.0%) | 0 (0.0%) | 3(0.0%) | 3(10.0%) |
| Professional experience(years) | < 1 | 5(16.4)% | 2 (6.7%) | 0(0.0%) | 7(23.3%) |
| | > 1 | 6(20.0%) | 13 (44.0%) | 4(13.7%) | 23(77.7%) |

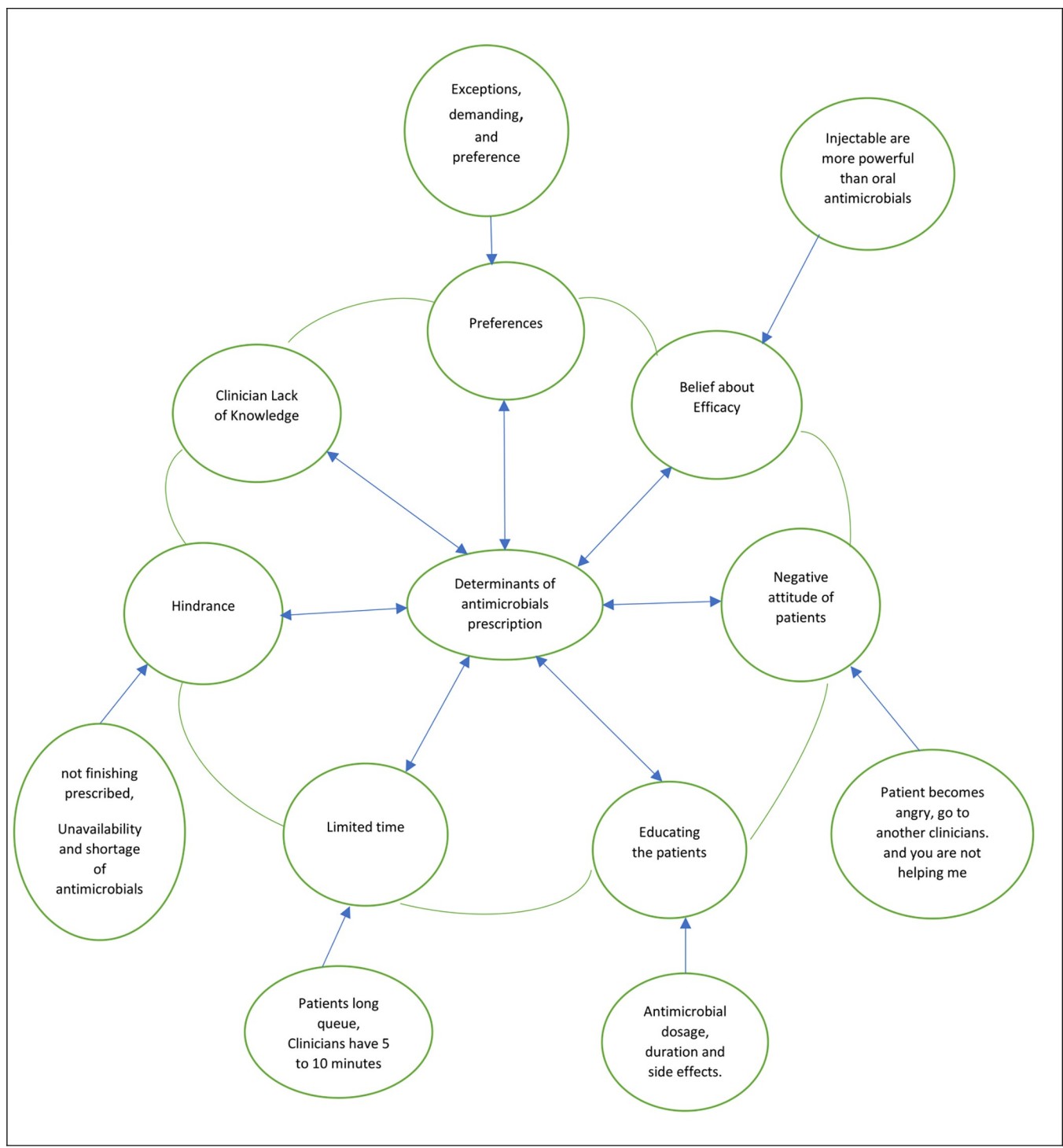

**Fig 1. Map of sub themes and major themes.**

*"Okay, it's about the patient preferences, okay, it's just like there are a lot of antibiotics but patients may choose that "I like this antibiotic when I take it, I feel good, I recover from my complaints and my disease", could be just patient expectation according to clinical condition of her disease" (Clinician # 5).*

> *"aah sometimes aah patient do have their preferences on which drugs antibiotics to be prescribed to them that they feel is best for them not the condition they have, what's best for them"* (Clinician # 6).

## Theme 2: Belief about efficacy

The clinical officers reported that patients wanted IM and IV injections, since most of them believe that intramuscular injections, IM and IV antibiotics work better than per os (PO). So, in this case, if you give them PO antibiotics, they believe that you have not helped them. However, they believe that they will dramatically change for the better within two days, if you give them antibiotics.

> *"Patients' understanding of antimicrobials is that the belief that antimicrobials, especially injectable heal any form of severe illness. Even if it is not a bacterial infection, they still think that if you give them IV [Intravenous] antimicrobials, they are going to recover"* (Clinician # 7).

Clinicians also reported that prescribing antibiotics occurs even in suspected cases of viral infections or mere cough. This was because patients and guardians believe their patients will improve only after taking antibiotics and that any form of illness can be cured with antibiotics. Thus, patients show lack of knowledge on antimicrobials that can be used to treat a bacterial infection or a viral infection.

> *"I think they feel that for them to get well then, they have to take a certain type of antimicrobial. Whether you find that the malaria test is negative then they still have the feeling that for them to get well they have to take antimalarial, or for them they just have a viral infection like cough or whatever or just a flu. They believe that for them to get well they have to take antibiotic like amoxicillin"* (Clinicians# 12).

Clinicians reported that patients believe that having a cough means one needs antibiotics. This belief prevents proper antimicrobial prescription in a health care setting.

## Theme 3: Negative attitude of patients towards clinicians who do not prescribe antimicrobials

Most of the clinicians reported that when they refuse to prescribe antimicrobials, patients have a negative attitude towards such clinicians. Clinicians reported they felt that patients had a negative attitude towards such clinicians. Almost all clinicians reported that such patients think of them as being incompetent.

> *"Well, when you refuse, they think maybe you are not a good clinician. You don't really know your work or else you haven't helped them. They would prefer to go to another clinician or else to go to another hospital where they feel they can be helped. They feel that you haven't helped them"* (clinician #12).

Almost all clinicians reported that such patients think of them as being incompetent when they refuse to prescribe antimicrobials to them.

> *"They do just think that am not a medical practitioner, that am not well equipped with knowledge and they go for another clinician"* (Clinician # 4).

## Theme 4: Educating the patients

Patient education is a factor that can lead to a reduction in unnecessary antimicrobial prescriptions. As one way of reducing inappropriate antimicrobial prescription, the majority of Clinicians reported the need for patients' education during patients' clinic visits. Clinicians suggested that the following should be done during clinician-patient interaction in health care settings: educating patients on the dosage, drug taking and period, any side effects as well as when to return to the hospital if there is no improvement. All these need to be properly communicated.

*"First, we need to explain why we are giving those drugs, frequency, route, and duration. And they should not share with someone else because it's only for him or them that have attended the service and the drug is prescribed only for him. I think about the problems people still share drugs somewhere behind. You should build a good relationship between you and the patient and make sure when you have given the drug they should come back for feedback" (Clinician #11).*

Patients' education should include topics on antibiotic and antimicrobial resistance. This important theme of educating the patients is however restricted by the reality on the ground as shown in the next theme.

**Theme 5: Limited time / clinicians being overwhelmed.** Clinicians stated that limited consultation time between clinician and patient was one of the factors that lead to inappropriate prescription of antimicrobial. It emerged that there is limited time a clinician can spend with an individual patient because of large numbers of patients visiting a health facility. This is a big challenge that affects clinicians. There is a lack of comprehensive history taking on patients because of having to spend minimal time with each one of them. Some clinicians may see as many as 50 patients per day.

*"We can spend 2 minutes with each patient because we have long lines in outpatient departments and sometimes there is one clinician or two, so if you take much time with patients, they start complaining that you are wasting their time. It affects a lot because we need to have more time with our patient and they should talk more of their complaints but with the complaints that I said that we are few Clinicians, we spend aah not enough time with the patient, so the patient do not share more of their complaints that they have come with on that particular day" (Clinician #1).*

The underlying factor is handling a long queue of patients in outpatient departments and huge workload. This results in unnecessary antimicrobials prescription in health facilities in order to relieve pressure in the outpatient department.

Lack of enough time is a barrier for proper history taking, physical examinations, investigations and counselling. Furthermore, because of having fewer clinicians, antimicrobials are being prescribed in order to attend to more patients within a short period of time. As a result there is no explanation to patients on what they are suffering from and the importance of adhering to medications.

## Theme 6: Hindrance / obstacle to antimicrobial prescription

The study also investigated problems of antimicrobial prescription in cases where this is the appropriate decision. Clinicians stated that they do face problems and challenges in prescribing some antimicrobials because of their unavailability in the health care settings. The findings

revealed that inadequate availability of antimicrobials and the same being sold on the open market are problems since people can go and buy without a prescription from a clinician.

*"One of the challenges mostly (silence) its aah repetitive usage of single antimicrobial, even in the same patients or in most outpatients seen or even inpatients, so it's mostly certain antibiotic dominate over other antibiotics, so that's one of the challenges simply because it has developed some resistance simply because of overuse and it has caused most of the unfavorable side effects and which are most difficult to treat so are some of the challenges we have met so far" (Clinician # 7).*

Clinicians reported that a shortage of certain antimicrobials makes them prescribe the same antibiotics, even in cases where they believe it is not the best option, not indicated or not the strongest one.

*"Your choices may be out of stock in a particular pharmacy and that can affect your prescription as well. And the other thing is, you are not quite sure what you are treating. So you just prescribe but then you are not really sure like blinded and treating blindly" (Clinician # 12).*

### Theme 7: Clinician lack of knowledge on antibiotic and antimicrobial resistance

Besides all that, participants showed that they had minimal understanding of antibiotics and their resistance. Few clinicians correctly defined what antibiotic and antimicrobial resistance mean. Clinicians were also using antibiotics and antimicrobial resistance interchangeably. In this study, clinicians did not define properly what antibiotics and antimicrobials are. Below is an illustration.

*Definition of Antimicrobial resistance: Antimicrobial resistance means the causative organisms, the bacteria have developed a mechanism or a resistance to that antimicrobial which means you might give antimicrobial which previously could work or the bacteria could respond or could be sensitive to that antimicrobial but now in the later stage or after a certain period of time the bacteria will develop another mechanism against that antimicrobial" (Clinician # 7).*

Although clinicians were aware of the causes of antimicrobial resistance, but they showed lack of knowledge on the proper definition of antibiotic resistance and antimicrobial resistance.

### Discussion

Since medical assistants, clinical technicians and clinical officers are front-line health workers in primary and secondary health facilities in Malawi. They are more likely to prevent inappropriate prescription of antimicrobials and educate patients if they have enough knowledge and are aware of antimicrobial resistance. This study yields important findings regarding factors influencing clinicians to give over their prescribing decision power to patients during consultation. The study identifies an important area that needs to be addressed when developing education interventions regarding interactions between clinician and patients. The study has also demonstrated that only few clinicians were aware on the definition of antibiotic and antimicrobial resistance. In a study done by Tam *et al.* [6] reported that over 2 decades, the resistance

of Gram-negative pathogens to all empiric, first-line antimicrobials, ampicillin/penicillin, gentamicin, ceftriaxone used in Malawi, is high, rising, and most marked among young infants [6].

The majority asserted that some factors influence clinicians to give over their prescribing decision power to patients. Key among the factors are preferences, beliefs and efficacy of antimicrobials, negative attitude of patients towards clinicians, limited time /Clinician being overwhelmed as well as hindrance / obstacle to antimicrobial prescription. However, there is a significant gap on the definition of antibiotic and antimicrobial resistance among Clinicians which needs to be bridged, and that can result into appropriate antimicrobial prescription.

## Preference

The present study confirmed that patient preference is a factor that influences clinicians to give over their prescribing decision power to patients and it determines inappropriate antimicrobial prescription in health care settings. These findings are supported by several studies in developed countries [11, 24–27]. This is also reflected in a similar study, in which, one of the reasons for the prescription of antimicrobials is patient demands or attitude [28]. A study done in Egypt also reported that preferences of caregivers and patients were among of the factors that contribute to antibiotics prescriptions [27].

## Belief about efficacy

Our study also found out that belief about efficacy among patients in antimicrobials is contributing to inappropriate prescriptions in health care settings. It is reported that patients have a belief in certain antimicrobials over others. Clinicians in this study cited that patients' demands and preferences for injectable or intravenous antimicrobials over oral ones contribute to inappropriate prescription. Similar findings from developed countries also reported a patients' demand for antibiotics to avoid repeated consultations [29–32].

The current findings are supported by another study which found that patients come to a hospital with a common cold and later demand intravenous antibiotics. It further asserts that patients believe that intravascular are better than oral antibiotics [27, 28, 32–34].

## Negative attitude of patients towards clinicians

Our findings about negative attitude of patients towards clinicians who refuse to prescribe antimicrobials is similar to other study findings which found out that clinicians were prescribing antibiotics in fear of losing patients' trust [35].

The results also show that patients would change physicians when antibiotics are not prescribed. This is also reflected in similar studies that reported that even when patients do not need medication, doctors prescribe antibiotics to maintain a good patient-doctor relationship [36–38].

## Educating patients

This study revealed that clinicians were influenced to prescribe antimicrobials because of patients' lack of education on medications. Clinicians reported that they do not have enough time to counsel and educate patients during consultations because of demands to attend to large patient numbers. In a review conducted by Ayukekbong et al. [39], it was found that, because of high patient-doctor ratio in most developing countries, doctors are overwhelmed. As a result, there is often inadequate time for meaningful education and communication with

the patient on drug adherence guidelines and consequences of poor or non-adherence to the guidelines.

Findings in this study are all consistent with other studies done in Belgium, England and France which reported that mass media interventions such as national TV campaigns and campaigns through other forms of mass media reduce antibiotic prescribing for Acute Respiratory Tract Infection. However, these strategies work best when targeting both healthcare professionals and the public [40].

## Limited—time / clinician being overwhelmed

The findings also reveal that limited time act as a barrier to proper antimicrobial prescriptions. Clinicians reported that they prescribe antimicrobials in order to handle long queues in the outpatient department. Several studies support the fact that clinicians spend less time with patients because of work overload.

It is reported that clinicians prescribe medications to end the consultation. They also reported that they prescribe under pressure when factors other than clinical presentations pushed them into prescribing even when they believe antibiotics are not needed [31]. In a study conducted in Karnataka state in South India, physicians agreed that they have too much work because of staff shortages and nearly half of them said that their patients ask them to prescribe antibiotics [41].

## Hindrance / obstacle to antimicrobial prescription

Several factors such as unavailability of antimicrobials or their shortage, as well as antibiotics being sold on the open markets are barriers to proper antimicrobial prescriptions since people can go and buy antimicrobials from pharmacies and open markets without a prescription from a clinician. One of the barriers to appropriate antibiotic prescriptions is inappropriate antibiotic use which has resulted from lack of access to and affordability of antibiotics due to inadequate government funding in developing countries [42].

A study done by Baubie, et al. [43] also reported that high physician workload and high antibiotic use in the community were major barriers to antimicrobial stewardship implementation and lack of patient/client understanding of antibiotics, and difficulty in making diagnoses were barriers to proper antimicrobial prescription [44].

The current study is also supported by another study done in India which shows that selection of particular antibiotics also depends on their availability at the public health center and this is a barrier to prescribers [41].

Similarly, a study done in South Asia reveal that common challenges to proper antimicrobials prescription were; poor dispensing, poor quality antibiotics, less effective ones in hospital, insufficient history taking and sale of antibiotics that have no proper dosage or are clinically inappropriate [45].

## Clinician lack of knowledge on antibiotic and antimicrobial resistance

Furthermore, we need to educate clinicians on antibiotic and antimicrobial resistance. Overall, participants had minimal understanding of antibiotics and antimicrobials resistance. In this study, clinicians pointed out that overuse, poor adherence, and self-medication were causes of antibiotic resistance. In a study done in France and Scotland, the clinicians had knowledge of antibiotics resistance [46].

In the current study, clinicians knew the causes of antimicrobial resistance and had knowledge which is similar to the findings of a study done in Saud Arabia on rural and urban physicians which pointed out that inadequate prescription, use of antimicrobials without

prescription and noncompliance of patients are the most important factors contributing to the development of bacterial resistance to antibiotics [46, 47].

**Strength of the study.** This is the first study done in Malawi among clinical officers on antimicrobial prescription. The study managed to capture a wide range of determinants of inappropriate antimicrobial prescription. Sample size in qualitative research is determined by data saturation and it is a gold standard in a qualitative study and (n = 30) were all interviewed. The study had high levels of participation which might show that research participants were interested in antimicrobial resistance and they were willing to participate in the study.

Limitation of this study was only done in one district, Mulanje, Southern Malawi, so it is a snapshot of Mulanje district as such the result cannot be generalized. Another limitation is non-random sampling. Finally, private clinicians were not interviewed which is also one of the limitations of the study.

## Conclusion

Based on the findings in this study, the following are key conclusions that contribute to the evidence of determinants of antimicrobial prescription. Patients or the general public need more sensitization of the AMR issue as a whole which is crucial and paramount to improve antimicrobial prescription. Doctors/ COs if well trained can educate the patients if they demand so. Future study could better evaluate the understanding of clinical officers on proper antimicrobial use and factors that can contribute AMR.

## Supporting information

**S1 Data. Research data.**
(XLSX)

**S1 Text. Research transcript.**
(DOCX)

**S2 Text. Research transcript.**
(DOCX)

**S3 Text. Research transcript.**
(DOCX)

**S4 Text. Research transcript.**
(DOCX)

**S5 Text. Research transcript.**
(DOCX)

**S6 Text. Research transcript.**
(DOCX)

**S7 Text. Research transcript.**
(DOCX)

**S8 Text. Research transcript.**
(DOCX)

**S9 Text. Research transcript.**
(DOCX)

**S10 Text. Research transcript.**
(DOCX)

**S11 Text. Research transcript.**
(DOCX)

**S12 Text. Research transcript.**
(DOCX)

**S13 Text. Research transcript.**
(DOCX)

**S14 Text. Research transcript.**
(DOCX)

**S15 Text. Research transcript.**
(DOCX)

**S16 Text. Research transcript.**
(DOCX)

**S17 Text. Research transcript.**
(DOCX)

## Acknowledgments

The authors acknowledge support from NORERD and are much grateful to all participants who took part to participate in the study, Director of Health and Social Services at Mulanje District Hospital and Director of Mulanje Mission Hospital for allowing us to collect data.

## Author Contributions

**Conceptualization:** Morris Chalusa, Felix Khuluza, Chiwoza Bandawe.

**Data curation:** Morris Chalusa.

**Formal analysis:** Morris Chalusa, Chiwoza Bandawe.

**Investigation:** Morris Chalusa.

**Methodology:** Morris Chalusa.

**Resources:** Morris Chalusa.

**Supervision:** Morris Chalusa, Felix Khuluza, Chiwoza Bandawe.

**Validation:** Morris Chalusa, Felix Khuluza.

**Writing – original draft:** Morris Chalusa.

**Writing – review & editing:** Morris Chalusa, Chiwoza Bandawe.

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
