## [Decision Letter · Decision Letter 0]

14 Jul 2022

PGPH-D-22-00767

Determinants of Clinician and Patient to prescription of antimicrobials:case of Mulanje

Dear Dr. Chalusa,

Thank you for submitting your manuscript to PLOS Global Public Health. After careful consideration, we feel that it has merit but does not fully meet PLOS Global Public Health’s publication criteria as it currently stands. Therefore, we invite you to submit a revised version of the manuscript that addresses the points raised during the review process.

We look forward to receiving your revised manuscript.

Kind regards,

Cemil Kurekci

Academic Editor

Journal Requirements:

1. Please amend your detailed online Financial Disclosure statement. This is published with the article. It must therefore be completed in full sentences and contain the exact wording you wish to be published.

State the initials, alongside each funding source, of each author to receive each grant.

2. In the Funding Information you indicated that no funding was received. Please revise the Funding Information field to reflect funding received.

Please ensure that the funders and grant numbers match between the Financial Disclosure field and the Funding Information tab in your submission form. Note that the funders must be provided in the same order in both places as well.

3. Please update your online Competing Interests statement. If you have no competing interests to declare, please state: “The authors have declared that no competing interests exist.”

4. In the online submission form, you indicated that “The datasets used and analyzed during the current study are available from the corresponding author on reasonable request.”. All PLOS journals now require all data underlying the findings described in their manuscript to be freely available to other researchers, either 1. In a public repository, 2. Within the manuscript itself, or 3. Uploaded as supplementary information.

5. Please ensure that the Title in your manuscript and the Title in your online submission form are the same.

6. Please assign a number, e.g. Fig 1, and include a legend for the figure found in page 7. Please also make sure the figure is cited or referred in the manuscript.

7. Please provide separate figure file in .tif or .eps format and remove any figures embedded in your manuscript file. Please also ensure that all files are under our size limit of 10MB.

8. We have noticed that you have uploaded Supporting Information files, but you have not included a list of legends. Please add a full list of legends for your Supporting Information files after the references list.

9. We appreciate you for providing a link to the source for the image found in page 7 (https://global-uploads.webflow.com/6061a9d807f5368139d1c52c/6107258d0fe57111f73f6428_Mulanje-SEP.pdf) in a separate correspondence. However, upon reviewing the document we cannot find any indication that the image is provided in an open source manner which would be acceptable for publishing under PLOS. 

If any of the images in the manuscript have been previously copyrighted, PLOS ONE is unable to publish these images, as all content is published under the Creative Commons Attribution (CC BY) 4.0 license, which means that they will be freely available online, and any third party is permitted to access, download, copy, distribute, and use these materials in any way, even commercially, with proper attribution.

We require one of the following:

- Specific consent from the copyright holder to publish these images in PLOS ONE, under the CC BY 4.0 license. Please include a file including the granted permission as a Supporting Information file and include the following text within your figure caption(s): “Republished from [ref] under a CC BY license, with permission from [name of publisher], original copyright [original copyright year].”

- The authors remove these images from their submission.

USGS National Map Viewer (http://viewer.nationalmap.gov/viewer/)

USGS Earth Resources Observatory and Science (EROS) Center (http://eros.usgs.gov/#)

The Gateway to Astronaut Photography of Earth (https://eol.jsc.nasa.gov/)

Maps at the CIA (https://www.cia.gov/library/publications/the-world-factbook/docs/refmaps.html)

NASA Earth Observatory (http://earthobservatory.nasa.gov/)

Landsat (http://landsat.visibleearth.nasa.gov/)

Natural Earth (http://www.naturalearthdata.com/)

Additional Editor Comments (if provided):

Reviewers' comments:

Reviewer's Responses to Questions

**Comments to the Author**

1. Does this manuscript meet PLOS Global Public Health’s publication criteria? Is the manuscript technically sound, and do the data support the conclusions? The manuscript must describe methodologically and ethically rigorous research with conclusions that are appropriately drawn based on the data presented.

Reviewer #1: Yes

Reviewer #2: Partly

2. Has the statistical analysis been performed appropriately and rigorously?

Reviewer #1: Yes

Reviewer #2: No

3. Have the authors made all data underlying the findings in their manuscript fully available (please refer to the Data Availability Statement at the start of the manuscript PDF file)?

Reviewer #1: Yes

Reviewer #2: Yes

4. Is the manuscript presented in an intelligible fashion and written in standard English?

Reviewer #1: Yes

Reviewer #2: No

5. Review Comments to the Author

Reviewer #1: I think this is a very important topic and glad to see this work being done. In particular, I am glad to see the focus on the CO angle and I would make this front and center.

I would try to shorten the paper, perhaps removing the discussion of many other locations but can cite these works as other examples of these concepts.

I would briefly place AMR in context. Although an emerging issue, AMR is already worse in terms of mortality in Africa than any other continent (study includes Malawi)

https://www.thelancet.com/journals/lancet/article/PIIS0140-6736(21)02724-0/fulltext#%20

I would also point to specific issues regarding AMR in this setting, such as specific issues regarding neonatal sepsis very much point to

https://www.thelancet.com/journals/laninf/article/PIIS1473-3099(21)00050-5/fulltext

I might make a brief comment in the first section that AMR is driven by more than provider antibiotic prescriptions (livestock, waste, waterways) and that even legitimate prescriptions can drive amr.

I would try to explain what infections as well as antibiotics and antimicrobials are commonly used, if can give any context.

I would try to cut down the first section to focus on the role and magnitude of CO prescriptions in the setting, what is known about antibiotic decision making, and AMR locally. I would also stress that this survey explores their attitudes and perceptions, might supplement with any formal KAPs that have been published and any more info about knowledge.

Line 93 -96 I would explain further that need CO and techs and med assistants because they prescribe antibiotics and provide an important amount of the antibiotics prescribed but have often not been the subject of study, which has focused on medical doctors. would explain how many COs there are, relative numbers. and any info on scope of practice and any quantification of role in prescribing abx. in many countries the numbers of this cadre of health providers means they are prescribing more abx but there is little focus on them.

93 Since a previous study on antimicrobial prescription

94 focused on physicians and Medical doctors, it was necessary to also get views from Clinical

95 Officers, Clinical Technicians and Medical Assistants, particularly in Sub Saharan Africa who are

96 at the frontline in providing health services in primary and secondary health care settings

102 AMR may also be driven by appropriate antibiotic prescription. I might word the concern to state that trying to understand the prescription patterns that can drive amr.

Line 109-110

can you give more info on the hospitals? rural, urban? catchment area? This can be said in conjunction with the map.

Line 131 can cut down on some: Analysis began soon after data collection to get familiarization, which involved reading the

132 transcripts repeatedly and noting down ideas.

I would see if it's possible to quantify any of the themes, whether something almost all felt, or if there were divisions.

When discussing Themes, would discuss quite factually without any assumptions.

For Theme 1 might say: The interviewer asked them about determinants of antimicrobial prescription in health care settings.

instead of

To explore why clinicians give over their prescribing decision power to patients, the interviewer

154 asked them about determinants of antimicrobial prescription in health care settings.

For Theme 2, I would stick to what the survey showed.

I would adjust the line to say the clinical officers reported that patients wanted IM and IV injections (if that is so). As the patients were not interviewed, would not say the patients wanted this or cite another source documenting this. Although we have all seen patients request injections, without documenting the preference, would need to focus on what the clinical officers said.:

Most patients believe that intramuscular injections, IM and IV antibiotics work better than per os

169 (PO), so, if you give them PO antibiotics, they believe that you have not helped them. They believe

170 that if you give them antibiotics they will dramatically change for the better within two days.

For Theme 3,I would keep to stating what the clinicians stated.

I would keep to line 188, "Most of the clinicians reported that when they refuse to prescribe antimicrobials..." but on line 190, would change "In other words, the patients have a negative attitude towards such

191 clinicians." to the clinicians reported they felt the patients had a negative attitude towards such clinicians...

Likewise for line 196 would change "Almost all clinicians reported that such patients think of them as being incompetent " to perhaps almost all clinicians reported that patient said they were incompetent or they felt the patients thought they were incompetent.

For Theme 4, I might clarify more about "Patient education is a factor that can lead to a reduction in unnecessary antimicrobial prescriptions.

214 This important theme of educating the patients is however restricted by the reality on the ground

215 as shown in the next theme." Education described as more about dosage and indications to return.

For Theme 7, I would possibly adjust the statement "Besides all that, participants showed that they had minimal understanding of antibiotics and

257 antibiotic resistance." and simply state very clearly what understanding was lacking to avoid anything that might be interpreted as judgement.

what was the definition that was acceptable? what errors were made?

Four clinicians correctly defined what antibiotic and antimicrobial resistance

258 is, which represents 13 %. 6.5 % correctly defined antibiotic resistance whilst only 3% correctly

259 defined antimicrobial resistance.

I would possibly say that future study could better evaluate the understanding of clinical officers of proper antimicrobial use and risk of amr or expand on what they did not show comprehension of. At times one may have an understanding without being able to define a term.So might adjust the sentence:

Although clinicians were aware of the causes of antimicrobial resistance, they showed lack of

429 knowledge on antimicrobial resistance. Lack of patient education, limited time and work overload

430 are among the factors that promote inappropriate prescription of antimicrobials.

Reviewer #2: This study provide some data, however its academic writing needs to be improved by native speaker, or English editing service. Major revision.

Please re-write the sentences Line 52-54, also Line 57-62.

Line 63-67 is also hard to understand.

Line 68-73 looks like a repetition of the previous paragraph.

Line 83. please change "too many" with more appropriate word.

Line 85. Please delete both "too"

Please do not write "Clinical Officers, Clinical Technicians and Medical Assistants, Medical" with capital.

Please re-write line 105-108.

Line 116-117 and Line 145-146 please do not use capital letter.

Please do not write occupations name with capital.

For results section, why not authors give the percentage of responses and analyze data for gender etc.

6. PLOS authors have the option to publish the peer review history of their article (what does this mean?). If published, this will include your full peer review and any attached files.

**Do you want your identity to be public for this peer review?** For information about this choice, including consent withdrawal, please see our Privacy Policy.

Reviewer #1: No

Reviewer #2: No

---

## [Decision Letter · Decision Letter 1]

29 Aug 2022

PGPH-D-22-00767R1

Determinants of Clinician and Patient to prescription of antimicrobials:case of Mulanje

Dear Dr. Chalusa,

Thank you for submitting your manuscript to PLOS Global Public Health. After careful consideration, we feel that it has merit but does not fully meet PLOS Global Public Health’s publication criteria as it currently stands. Therefore, we invite you to submit a revised version of the manuscript that addresses the points raised during the review process.

We look forward to receiving your revised manuscript.

Kind regards,

Cemil Kurekci

Academic Editor

Journal Requirements:

1. Please ensure that the Title in your manuscript and the Title in your online submission form are the same.

2. Please ensure that you refer to Table 1 in your text as, if accepted, production will need this reference to link the reader to the table.

Additional Editor Comments (if provided):

Reviewers' comments:

Reviewer's Responses to Questions

**Comments to the Author**

1. If the authors have adequately addressed your comments raised in a previous round of review and you feel that this manuscript is now acceptable for publication, you may indicate that here to bypass the “Comments to the Author” section, enter your conflict of interest statement in the “Confidential to Editor” section, and submit your "Accept" recommendation.

Reviewer #3: (No Response)

Reviewer #4: (No Response)

2. Does this manuscript meet PLOS Global Public Health’s publication criteria? Is the manuscript technically sound, and do the data support the conclusions? The manuscript must describe methodologically and ethically rigorous research with conclusions that are appropriately drawn based on the data presented.

Reviewer #3: Yes

Reviewer #4: No

3. Has the statistical analysis been performed appropriately and rigorously?

Reviewer #3: N/A

Reviewer #4: N/A

4. Have the authors made all data underlying the findings in their manuscript fully available (please refer to the Data Availability Statement at the start of the manuscript PDF file)?

Reviewer #3: Yes

Reviewer #4: No

5. Is the manuscript presented in an intelligible fashion and written in standard English?

Reviewer #3: Yes

Reviewer #4: No

6. Review Comments to the Author

Reviewer #3: See attachement.

Reviewer #4: (No Response)

7. PLOS authors have the option to publish the peer review history of their article (what does this mean?). If published, this will include your full peer review and any attached files.

**Do you want your identity to be public for this peer review?** For information about this choice, including consent withdrawal, please see our Privacy Policy.

Reviewer #3: No

Reviewer #4: No

---

## [Editor Report · Decision Letter 2]

21 Oct 2022

Determinants of Clinician and Patient to prescription of antimicrobials:case of Mulanje, Southern Malawi

PGPH-D-22-00767R2

Dear Mr Chalusa,

We are pleased to inform you that your manuscript 'Determinants of Clinician and Patient to prescription of antimicrobials:case of Mulanje, Southern Malawi' has been provisionally accepted for publication in PLOS Global Public Health.

Best regards,

Cemil Kurekci

Academic Editor